# Climate Policy in Developing Countries: Analysis of Climate Mitigation and Adaptation Measures in Egypt

**Hamdy Abdelaty [1,2,\*], Daniel Weiss [1] and Delia Mangelkramer [1]**

1   Management Department, School of Business and Economics, Freie Universität Berlin, 14195 Berlin, Germany
2   Business Administration Department, Faculty of Commerce, Cairo University, Giza 12613, Egypt
\*   Correspondence: hamdy.abdelaty@fu-berlin.de

**Abstract:** The Nationally Determined Contribution (NDC) refers to a country's climate action plan to limit greenhouse gas (GHG) emissions and adapt to climate change hazards. Each country is obliged to submit its NDCs to the UNFCCC, adhering to a guideline for increasing clarity and transparency. Nonetheless, few studies have employed this guideline to assess countries' contributions, particularly the NDCs of developed countries. Our article centers on the case of The Arab Republic of Egypt (hereafter Egypt), which is extremely susceptible to climate change impacts due to its geographic location and economic structure. Using desk research and a systematic NDC analysis, this paper reviews recent measures Egypt has taken to build national resilience against climate change. We also assess Egypt's planned mitigation and adaptation measures until 2030, documented in its updated NDC according to four criteria: mitigation ambition level, comprehensiveness, implementation plan, and transparency. The results show that Egypt's 2022 NDC is more advanced on different fronts than the 2015 submission, focusing on fewer sectors and specific quantified targets for mitigation and adaptation. However, the updated NDC only partially meets the essential criteria for mitigation ambition level, implementability, and transparency. We provide a set of methodological and policy recommendations for improvement.

**Keywords:** climate policies; NDCs; mitigation and adaptation; developing countries; Egypt

## 1. Introduction

The adoption of the Paris Agreement (PA) in 2015 was a milestone in the history of international climate negotiations. As part of the agreement, countries are mandated to submit their Nationally Determined Contributions (NDCs) as a landmark adoption of the agreement and central component of its implementation [1]. An NDC reflects the country's ambition to limit national greenhouse gas (GHG) emissions and build resilience to adapt to the impacts of climate change [2]. Recent research underscored that if countries' ambition is not increased before 2030, global mean temperature increase can no longer be limited to 1.5 °C above pre-industrial levels [3–6]. Recognizing the urgency of the challenge, the Glasgow Climate Pact also encouraged countries to submit stronger 2030 emissions reduction targets to move steadily toward achieving the global aim [7]. Therefore, countries are requested to update their NDCs every five years and increase their ambitions with every cycle [2]. The first round of NDCs was submitted in 2015 as intended contributions. Recently, countries have submitted or are finalizing their updates for the new cycle of submissions in 2020/2021. A total number of 194 first NDCs and 11 s NDCs have now been recorded in the NDC registry [8].

While some previous research analyzed national strategies and resilience plans against climate change [9–11], a wide range of scientific work focused particularly on NDCs [12–14]. However, only a few studies have adopted a holistic approach in assessing and capturing the progress in the new submissions of the nations' NDCs (2020/2022) in comparison to the first submission (around 2015) and UNFCCC's guideline information to facilitate Clarity,

Transparency, and Understanding of NDCs (ICTU) [15]. Moreover, the above issues have been well recognized, but the literature has paid scant attention to developing countries, especially in the African case.

We fill this gap by analyzing the case of Egypt as one of the most vulnerable developing countries to climate change, primarily due to its geographical typology and dependence on the Nile River [16–18]. Various studies have recognized this vulnerability and investigated climate change efforts in specific sectors, such as water resources [19], energy provision [20], and the food system [21]. However, a more comprehensive assessment of Egypt's climate change efforts is still missing. Before these research gaps, our study aims to answer the following research questions: What are the main climate adaptation and mitigation measures Egypt has applied and intends to apply until 2030? Are those measures ambitious enough and implementable? We aim to provide a detailed assessment of the Egyptian case and define the main blind spots in the current endeavors opening more windows for improvements for developing countries in general.

This study contributes to these questions utilizing comprehensive desk research and NDC analysis, following a range of studies in the scientific community proposing NDC assessment methods [22–24], and analyzing countries' NDCs to critically evaluate their content and climate policy implications [24–26]. In particular, we use the methodology proposed by the New Climate Institute and Öko-Institut e.V. in Germany, which has been recently developed to provide a more comprehensive, comparable, and replicable NDC analysis than previous assessment tools [24]. Herewith, we assess the NDC's mitigation ambition level, comprehensiveness, implementability, and transparency to contribute to the ongoing efforts of enhancing NDCs to achieve the goals of the Paris Agreement. Our analysis has the potential to inform policymakers, researchers, and other stakeholders on how Egypt can strengthen its climate change mitigation and adaptation efforts, particularly in light of its vulnerability to the impacts of climate change as a developing country.

The rest of the paper is structured as follows. Section 2 outlines Egypt as our research setting, discussing the ongoing climate policies, programs, and actions taken by the Egyptian government or by the international actors to support climate adaptation on different levels. Section 3 explains the assessment method, and Section 4 outlines the principal analysis and results. Sections 5 and 6 provide a detailed assessment of Egypt's updated NDC, followed by extended discussion and policy recommendations from policymakers.

## 2. Theoretical Background: Nationally Determined Contributions

The introduction of NDCs has sparked a wide range of associated academic works covering various related aspects of global climate politics [12–14]. First, a range of quantitative studies concerned themselves with the aggregated emission targets put forward in the different NDCs and whether they align with the goals of the Paris Agreement [27,28], repeatedly emphasizing the urgency to strive for more ambitious climate policies (e.g., [29–31]).

Second, scholars discussed human rights and political issues [32,33] as well as broader governance mechanisms related to NDC implementation, monitoring, and evaluation [34,35]. Relatedly, there is emerging work on the conditionality of NDC implementation, e.g., concerning receiving climate finance, technology transfer, and capacity building [1,36–39].

Third, a further strand of the literature examined factors that influence the ambitiousness of a country's climate policy, highlighting factors such as climate change vulnerability, local air pollution, fossil fuel dependency, energy independence, and economic development [40–42]. Fourth, there also has been focused work on reviewing individual NDCs [24–26] and even specific sector targets such as energy, transport, or agriculture [20,21,43]. Methodologically, these works have been accompanied by studies proposing different NDC assessment methods [22–24] and analyzing countries' NDCs to critically evaluate their content [24–26]. Recently, more and more scholars emphasized the climate politics of developing countries and re-sparked the decade-old debate about climate justice. Accordingly, there is a wide range of evidence highlighting that developing countries have to burden the greatest impacts of climate change, while developed countries carry the

greatest responsibility of causing global climate change [44–47]. Relatedly, studies show that poorer and more vulnerable countries tend to set less ambitious emission reduction targets, while rich and energy-deficient countries are more likely to set aggressive climate policies and ambitious emission reduction targets.

However, climate change has a direct impact on the lives and economies of countries, especially those in Africa and Asia that are dependent on agriculture and fossil fuels [42,48]. By the same token, the targeted development goals in developing countries often conflict with emission reduction goals because affordable and low-cost fossil fuel energy is necessary for solving social problems such as poverty and under-education [49]. At the same time, as mentioned above, the level of economic development plays a large factor in determining a country's ability and willingness to support and implement climate policies. Thus, measures to reduce GHG emissions and environmental degradation have also become a tool for addressing development challenges rather than just climate change [28,42].

## 3. Research Setting

### 3.1. Egypt's Economic Sectors and GHG Emissions

Egypt's historical contribution to global emissions is relatively low, estimated at 0.6%. Although a relative decoupling has been achieved, with GHG emissions growth lower than GDP growth, both remain closely linked, with positive emission growth in the last 30 years [50]. Egypt's emissions have trended upward between 1990 and 2019, both in absolute terms (163% increase) and per capita terms (47% increase) [51]. For instance, in 2015, the energy sector (including transport and industry emissions) and industrial processes accounted for 77% of emissions [50]. In 2019, the energy, transport, and industry sectors contributed over 80% to total emissions, with 74% from energy and 5.4% from industrial processes. Electricity and transport are the only sectors with an upward trend in emissions. Within the energy sector, natural gas and crude oil production and consumption remain the main sources of GHG emissions and primary energy supply.

Despite ambitious targets in Egypt's Integrated Sustainable Energy Strategy (ISES) 2035, natural gas and oil still represented 92% of the total primary energy supply in 2019. The industry sector, a key contributor to Egypt's GDP, is the third largest energy consumer at 28%. The iron and steel, aluminum, cement, and oil refinery sectors, which together emit 29 MtCO2e annually (9% of total GHG emissions in 2015), are priority sectors for greening production processes and energy sources [50,52]. Emissions in these sectors are expected to continue increasing if no action is taken. Transport, the fastest-growing industry, significantly contributes to air pollution [53]. Cairo's emissions are among the highest among global cities, leading to a health toll. Egypt has the highest morbidity and mortality rates due to ambient air pollution in the MENA region. In 2019, 150 per 100,000 people died prematurely due to air pollution [54].

The pressure on services such as electricity, water, food, health, and education will continue to increase due to Egypt's rapidly growing urban population. The United Nations estimates that by 2050, 55.6% of the population will live in cities [55]. A significant proportion of the country's urban population is exposed to climate hazards, such as flooding, heat stress, air pollution, and desertification. Egypt's direct CO2e emissions mainly come from cities and have increased tenfold in the past 50 years [52]. The projected temperature increase will worsen urban heat island effects and air pollution, decreasing livability and productivity. Unplanned and unregulated urban expansion is causing desertification and reducing arable land. Limited public transportation, green spaces, and social infrastructure, combined with heavy reliance on fossil fuel vehicles, lead to congestion and emissions. Only 1.7% of generated wastewater is recycled, contributing to water pollution [56].

### 3.2. Research Method

In order to capture the recent progress Egypt has made to tackle the impacts of climate change, we relied on intensive desk research investigating published material in English and Arabic including recent studies, international reports, Egypt's national strategies, new

regulations, and ministerial decrees. In order to assess what Egypt intends to do until 2023, we used a systematic method explained below to analyze Egypt's NDCs.

NDC Assessment Method

Countries' ambition alone is not enough. To realize their ambitions, NDCs should be underpinned with effective plans and measures to ensure the transparency and implementability of NDCs [1]. In the first round of submissions, intended NDCs were highly incoherent, demonstrating the difficulty of tracking the progress or comparing between Parties over time. Therefore, guidance for communicating NDCs was adopted at the climate change conference in Katowice in 2018, laying out the information to facilitate Clarity, Transparency, and Understanding of NDCs (ICTU) [15]. Parties also adopted guidelines for the transparency framework in Decision 18/CMA.1 [57], which addresses the information necessary to track progress in implementing and achieving NDCs.

Coinciding with the second round of submissions in 2020, various tools were developed to assess submitted NDCs and compare their components using different indicators that operationalize the ICTU guideline. These include the CAT Climate Target Update Tracker [22], the Climate Transparency NDC Transparency Check [58], the ClimateWatch 2020 NDC Tracker [23], Deutsches Institut für Entwicklungspolitik NDC Explorer [59], and the WWF #NDCsWeWant [60]. However, these assessment tools vary in their comprehensiveness, criteria, and replicability, prohibiting a consistent analysis and comparison of different NDCs. To address this gap, the New Climate Institute and Öko-Institut e.V. in Germany build up on the previous assessment tools to develop a systematic methodology, which we apply in this study. It provides a suitable tool for systematic analysis and comparison of NDCs over different countries and time periods with maximum transparency and replicability [24]. In doing so, the authors derived four comprehensive criteria that are essential for an NDC's success and cover the critical aspects of previous assessment tools as much as possible:

- Mitigation ambition: The NDC submission cycles urge countries to present updated NDCs that exceed their previous targets, primarily emphasizing enhancing mitigation efforts. If countries fail to increase their level of ambition while presenting a new NDC, then the cycle falls short of achieving its intended objective.
- Comprehensiveness: This element ensures the inclusion of all necessary NDC components, both mandatory and non-mandatory. The NDC should address climate change mitigation, all applicable emission gas types, and sector coverage on both mitigation and adaptation fronts.
- Implementation plan: The purpose of NDCs is to encourage Parties to undertake climate action at the national level. NDCs must be firmly based on the national policy-making process for successful implementation. A successful implementation is facilitated when certain conditions are met, including the existence of sector-specific objectives, a rigorous preparation process, a well-defined implementation plan, and the incorporation of the targets into the national laws of the country.
- Transparency: For Parties to voluntarily increase their NDCs, trust plays a crucial role. Such trust can be established through substantial transparency in the information disclosed in their respective NDCs. This includes transparent and comprehensible information on accounting methods and data, specifically concerning the resulting emission levels.

These four criteria are used to summarize the most significant findings of the NDC analysis and provide an effective overview for NDC comparisons. The ratings in the overview table are based on the detailed analysis of five NDC elements, namely scope, mitigation ambition, mitigation completeness, mitigation implementation, and transparency. For reasons of space, we refer readers to [24] (pp. 61–72), who provide a comprehensive understanding of how to evaluate each element. The structure of the analysis structure is intended to assist users in navigating the analysis and accessing the sections that interest them most. Each element is evaluated on a simple four-point rating system:



- + The new NDC provides more detail than the previous one.
- (-) The new NDC provides less detail than the previous one.
- O There is no change between the NDCs.
- ? The change between the NDCs is not clear.

In the following section, we describe the main results of our Egypt NDC of the four key criteria, which build up on the more detailed analysis of the five NDC elements that is reported in Appendix A.

## 4. Results: Recent Progress

### 4.1. Egypt's Institutional Arrangement and Policy Instruments

4.1.1. Climate Policy Actors

Egypt responds to climate change through a collaborative and multi-sectoral approach, in which many ministries work together and where the areas of responsibility intersect. At the ministerial level, the Ministry of Environment and the Ministry of Electricity and Renewable Energy are among those with the greatest institutional capacity. Central ministries include the Ministry of Water Resources and Irrigation and the Ministry of Environment, Housing, Utilities, and Urban Communities. The Ministry of Agriculture and Land Reclamation also undertakes the coordination of public institutions. The Ministry of Trade and Industry targets partnerships from the public and private sectors, including governments, non-governmental organizations, multi-lateral partners, such as funds, programs, and specialized agencies, as well as partners from academia, the business sector, and regional agents. Herewith, complementary expertise and shared values, aiming for a more inclusive, sustainable, and prosperous future, are the main drivers for new collaborations.

4.1.2. Climate Policy Strategies

Egypt's climate policy has evolved from following international commitments to a long-term strategy to become a regional leader in addressing climate change. Egypt has also established national strategies, made commitments, and set up steering mechanisms for climate change since joining the United Nations Framework Convention on Climate Change in 1994. Egypt launched its intended NDC in 2015 [61,62] and the updated NDC in 2022 [63], covering 2015–2030, and has expressed its commitment to a sustainable future and a just transition. The National Climate Change Strategy 2050, launched in 2022, provides a comprehensive framework for climate action to 2050 [64] with goals on mitigation and adaptation and aims to overcome governance, financing, technology, and awareness constraints. The strategy includes establishing specialized units in all ministries to foster coordination and has 26 high-priority projects to be completed by 2030 [64].

Egypt's commitment to tackling climate change challenges is augmented by being the first country in Africa to host the Conference Of Parties (COP27) in Sharm El-Sheikh in November 2022 [65]. Under the presidency of Egypt, Parties could reach a significant milestone in climate negotiations by agreeing on establishing the "Loss and Damage" Fund to translate Article 9 of the Paris Agreement 2015 into practice [18]. Accordingly, developed countries will allocate money to this fund to finance climate mitigation and adaptation programs in developing countries [2] (Article 9). One of the most recent and comprehensive programs in line with Egypt's National Strategy on Climate Change 2050 is the Country Platform for the Nexus of Water-Food-Energy (NWFE) program, initialized at COP27 [66]. It aims to mobilize international financial and technological support through a multi-stakeholder approach to address the interlinked energy, water, and food risks and integrates nine high-priority climate action projects for adaptation and mitigation.

4.1.3. Climate Adaptation and Mitigation Measures and Achievements

Egypt has implemented various climate policies striving for the mitigation and adaptation goals of Vision 2030. The following briefly overviews the most significant measures and achievements in relation to the first and second NDCs considering energy, agriculture and livelihoods, urban development, and the green finance Egypt NDC [61,63].

Egypt has undertaken comprehensive energy policy reforms to transform the dominating oil and gas sectors and build up a sustainable energy production. In doing so, the Ministry of Petroleum and Mineral Resources (MOPMR) established an institutional framework for energy efficiency in the petroleum sector, including the creation of a Higher Energy Committee and an Energy Efficiency and Climate Department [67]. The reforms reduced fossil fuel subsidies from 6% of the country's GDP in 2012/2013 to 0.3% in 2019/2020 [68]. At the same time, the government has encouraged investments to increase the electricity supply generated from renewable energy by 20% by 2022 and 42% by 2035. In this line, various policy measures in the ISES 2035 [69] have been implemented and resulted in a 340% increase in the total installed wind and solar power plants (5848 MW) from 2015/2016 to 2019/2020. Significant renewable energy projects include the Benban Solar Park (1465 MW), Assuit Hydropower Plant (32 MW), Kom Ombo Solar PV Plant (26 MW), and Gabal El-Zeit Wind Power Plant (580 MW) [70]. Concerning further energy efficiency improvements, the MOPMR began to follow ISO 50001 requirements and signed up to the World Bank Zero Routine Flaring Initiative to improve fossil fuel efficiency while reducing GHG emissions. Prominent programs include Improving the Energy Efficiency of Lighting and Building Appliances (2010–2017) [71], the Industrial Energy Efficiency (IEE) Project (2013–2018) [72], Solar Heating in Industrial Processes (SHIP) (2014–2022) [73], the Egyptian Programme for Promoting Industrial Motor Efficiency [74], the Environment Pollution Abatement Project Phase III (2017–2022) [75], the Within Hayah Karima National Initiative (Decent Life in English) [55], and the Household Natural Gas Connection Project [76], to provide households with improved access to a reliable, low-cost, and grid-connected supply of natural gas.

Agriculture and Livelihoods

Egypt is implementing various measures to modernize its agriculture and improve climate resilience and livelihoods. This includes a program to modernize irrigation systems with simultaneous restrictions on water-intensive crops and land reclamation projects. Further projects include the Sustainable Agriculture Investments and Livelihoods Project (SAIL) (2014–2023) and Building Resilient Food Security Systems to Benefit the Southern Egypt Region (2013–2018) [77]. Complementary water management efforts include extending and rehabilitating canals, building protection structures, and investing in desalination and aquaculture [78]. This also accounts for coastal and aquaculture adaptation projects, such as the Integrated Coastal Zone Management (2009–2017) under the responsibility of the Ministry of Water Resources and Irrigation (MWRI), and Enhancing Climate Change Adaptation in the North Coast and Nile Delta Regions in Egypt (2018–2024) [79]. The Emergency Food Security and Resilience Support Project is a response to the war in Ukraine, which has led to supply shortages for wheat. The project aims to mitigate supply and price shocks by mobilizing short-term emergency aid and strengthening food security and resilience in Egypt in the long term. This is intended to support particularly vulnerable groups [80]. In addition, the project also contributes to national climate change mitigation efforts by increasing the resilience of agriculture.

Furthermore, Egypt's vision statement includes investment in the Egyptian people, and it is pursuing this vision with reforms in the health sector to boost living standards. The project Transforming Egypt's Healthcare System is still an integral part of these reforms and will expire at the end of 2023 [81]. It has been improving the quality of primary and secondary health care services, enhancing demand for health and family planning services, supporting the prevention and control of Hepatitis C, and providing an immediate and effective response to an eligible crisis or emergency.

Infrastructure

The Egyptian government has also invested in its infrastructure, focusing on low-carbon transport and solid waste management. The Egyptian transportation sector is moving towards a low-carbon future with the recent opening of stage 4 of the third Cairo

metro line [82]. A new high-quality bus system, along with the development of electric ve-hicle (EV) infrastructure and an EV charging tariff [83], has been introduced to encourage a shift towards low-carbon mass transit. Expanding this transportation network is part of the more significant effort to reduce vehicle emissions and improve air quality in Greater Cairo, as outlined in the Air Pollution Management and Climate Change Project (2020–2026) [84]. The latter also aimed at improving solid waste management, going hand-in-hand with issuing the Waste Management Regulation Law 202/2020 and its executive regulations [84].

Further efforts include developing waste-to-energy and recycling facilities, landfill development, and collection and transportation systems for managing municipal solid waste. Other projects targeting climate mitigation through the degradation of pollutants are the Sustainable Persistent Organic Pollutants Management Project for the environmentally sound management and disposal of targeted stockpiles of obsolete pesticides [85] and the Pollution Management and Environmental Health Program to provide greater support for pollution management [86]. Here, particular emphasis is placed on protecting human health and economic growth. Egypt's Upper Egypt Local Development Program in the south of the country includes programs aimed at improving the management of municipal solid waste, conducting assessments of climate risks, and formulating local climate action plans to manage and reduce the impact of such risks on the local population and economy [87].

Egypt is also investing in improving the safety and service quality of rail services along the Alexandria–Cairo–Nag Hammadi corridor. The Railway Improvement and Safety for Egypt Project also targets climate change mitigation and adaptation for rural and disadvantaged populations [88]. The focus is on providing access to better, more affordable, environmentally friendly public transport. Other logistics projects include the construction of bypasses—for example, alternative routes for the congested Greater Cairo Area [89]. The main goal of the Cairo Alexandria Trade Logistics Development Project, for instance, is to decarbonize and improve the performance of the logistics and transport sectors in the Alexandria–6 October–Greater Cairo Area railway corridor.

Green Finance

Further climate efforts include Egypt's green finance efforts through various initiatives and mechanisms, including the launch of its first Sovereign Green Bonds in September 2020, worth USD 750 million [90]. Egypt's eligible green projects are worth USD 1.9 billion and are distributed among renewable energy (16%), clean transportation (19%), sustainable water and wastewater management (26%), and pollution reduction and control (39%) [91]. Implementing the Environmental Sustainability Criteria Guideline has increased green investments from 15% to 30% in 2020/21, with plans to reach 50% in 2024/25 [63] (p. 11). Financial regulatory authorities have also imposed Environmental, Social, and Governance (ESG) disclosure requirements on companies in the Egyptian Stock Exchange and those in the non-banking sector. In addition, attractive green financing is available through public and private financial institutions, including the Green Economy Financing Facility (GEFF).

*4.2. Results of NDC Analysis*

4.2.1. Mitigation Ambition Level

While Egypt's intended NDC did not encompass an emissions mitigation target at all, the new 2030 target is a relative improvement. The new submission identified quantified GHG emission reduction targets below Business As Usual (BAU) GHG emissions by 2030 for three sectors (i.e., electricity generation, transmission, and distribution; oil and gas; and transport, by 33%, 65%, and 7%, respectively, using 2015 as a reference year). The updated NDC also included a list of descriptive measures to mitigate emissions from other sectors. However, due to the lack of transparency, neither tracking the progress of the total emission mitigation nor judging the ambition level is possible. The updated NDC does not elucidate how much the total projected GHG emission reduction or the total GHG emission of Egypt will be in 2030 compared to the reference year of 2015 (325,515 GgCO2e) [50]. The Climate Action Tracker estimated that Egypt would need to commit to reducing its emissions by

approximately 25% by 2030 compared to today's level [92]. Moreover, Egypt does not have a national net-zero target to be achieved by 2050 or unconditionally identified emission reduction and mitigation targets. In sum, the NDC's mitigation ambition level component is underdeveloped.

### 4.2.2. Comprehensiveness

Comprehensives focus on the scope of the NDC regarding coverage of gas emissions, sectors, and adaptation. On the one hand, Egypt's intended NDC 2015 addressed more sectors with mitigation and adaptation actions than the updated NDC 2022 (i.e., mitigation, eight sectors vs. seven, and adaption, six sectors vs. four). On the other hand, the intended NDC included none of the seven gases to be reported under the Paris Agreement. In contrast, the updated NDC incorporated the three types that are relevant for developing countries (relevant GHGs here are defined for developing countries as the first three: Carbon Dioxide ($CO_2$), Methane ($CH_4$), and Nitrous Oxide ($N_2O$)), namely Carbon Dioxide ($CO_2$), Methane ($CH_4$), and Nitrous Oxide ($N_2O$). It is worth noting that the updated NDC explained why Egypt focuses its mitigation actions on three sectors (i.e., electricity generation, transmission, and distribution; oil and gas; and transport), as they account for 43% of the total emissions [63]. However, 57% of the total GHG emission remains opaque and untraceable. The new NDC cited that "*All Major sources of GHG emissions in the GHG inventory have already been covered in this NDC update*" (p. 37). However, it did not explain why mitigation actions did not cover critical sectors such as agriculture and land use (for developing countries with small emissions "main sector(s)" is defined as the sector(s) with the largest source of emissions; very often this only includes the energy sector and/or agriculture.). The agricultural sector is known as the second source of GHG emissions after energy, primarily associated with human activities such as burning fossil fuels, livestock, land use changes, fertilization, etc. [93].

Finally, both NDCs included a comprehensive section describing measures planned for building adaptation resilience in four common sectors: agriculture, water resources and irrigation, coastal zones, and urban development and tourism. However, the updated NDC provided a more concrete description of the planned adaptation actions with quantified targets than the intended NDC [62]. For instance, for creating agricultural climate resilience, measures such as rehabilitating 20,000 km of irrigation canals for water conservation and increasing the efficiency of current agricultural water use by 20% are clearly outlined. To build resilience in urban development and tourism, Egypt intends to maintain and expand the protectorates to cover 17% of the national marine and wildlife areas, with at least 5% constituting coastal areas. Yet, many other adaptation measures are generally described, and a lot of information is missing to judge whether these targets are ambitious or not, especially considering that the first version of the NDC lacks any quantitative targets. In sum, the comprehensiveness components of the NDC are adequately fulfilled.

### 4.2.3. Implementation Plan

The purpose of NDCs is to commit Parties to national action on climate. Therefore, NDCs must be rooted in national policy-making to ensure successful implementation. In this line, it is unclear whether Egypt's updated NDC or the updated mitigation targets are enshrined in national law. In addition, no implementation or action plans are presented in the NDCs despite mentioning that all planned measures are aligned with long-term national strategies. Although the updated NDC stated that the NDC preparation followed a participatory approach, no description of the preparation processes is included. Finally, implementing adaptation and mitigation actions is contingent on receiving technological and financial support from international actors. In sum, the implementability component is the weakest part of the updated NDC and is wholly unfulfilled.

### 4.2.4. Transparency

Transparency is evaluated in terms of (a) the availability of information to estimate the resulting emission of targets; (b) the clarity of the methodological approach following the ICTU guideline; (c) the transparency of the accounting modalities; and (d) the explanation of why the target is a fair contribution towards the global goal. While all items were nonexistent in the intended NDC, they have been marginally addressed in the updated NDC. There is a lack of transparency in disclosing the information required to estimate the resulting emission of the targets. There are also unclear planning processes and no quantitative explanation regarding the fairness of Egypt's contribution. The transparency component of the NDC is partially fulfilled.

## 5. Discussion and Policy Implications

Egypt is the world's 28th largest GHG emitter, with a total share of 0.71% of global emissions [94]. Despite this relatively marginal contribution of Egypt, its vulnerability to the impacts of climate change is critical [61,63]. The Nile Delta is very susceptible to sea level rise, and the rise of sea level by 1.00 m by 2100 would sink many coastal zones [95]. Moreover, the expected floods, drought, and erosion of the Nile Delta would impose severe conditions on the agriculture sector and risk food security in Egypt by 2030 [63]. These facts overshadowed the preparation of the NDC, envisaging more comprehensive adaptation measures than focusing on emission mitigation actions. This approach is not enough to solve the roots of the problem; instead, it handles the symptoms. For instance, Egypt's current policies would result in emissions reductions well below its target but only in line with a 3 °C warming [96]. Therefore, we outline the following recommendations to be considered in designing the successive NDC and to increase Egypt's resilience to climate change impacts.

### 5.1. Mitigation Progress and Transparency

While the previous NDC did not include an emissions reduction target, the new 2030 target is barely an improvement. However, tracking the progress of the total GHG emission mitigation in Egypt is not possible now and will not be possible in the future except for the three chosen sectors (i.e., electricity, transport, and oil and gas). It is unclear if the total emissions will continue increasing in an absolute term or above expected with the currently implemented policies. At the least, Egypt's GHG inventory should consider running such kind of calculations on the national or sectoral level for concrete planning. However, according to the Climate Action Tracker [92], Egypt would need to commit to reducing its emissions by approximately 25% by 2030 compared to today's level. The successive NDC should be more transparent and provide more details regarding the accounting modalities.

### 5.2. Mitigation Coverage: Inclusion and Exclusion

According to the ICTU guideline, Parag. 31(d) says that "Parties shall provide an explanation of why any categories of anthropogenic emissions or removals are excluded" [57]. Egypt's updated NDC stated that "*All Major sources of GHG emissions in the GHG inventory have already been covered in this NDC update*". The document explained why Egypt focuses its mitigation actions on three sectors, as they contribute 43% of the total emissions [63] (p. 37). Still, 57% of the total GHG emissions remain opaque and untraceable. The NDC also lacked transparency in citing which sources have been covered, which are excluded, and, most importantly, why.

### 5.3. Net-Zero Target Is Missing

Egypt also does not have a national net-zero target to be achieved in the second half of the century. Egypt's susceptibility to the risks of climate change necessitates a leading

role in defining a national net-zero target and encouraging other countries to adopt it during the next Conference of Parties, COP28. In addition, the urgency of the climate challenge makes the activation of Article 6 in the Paris Agreement regarding the creation of an emerging international carbon market imperative. Egypt expressed interest in voluntary cooperation in such a market; however, explicit actions and application instruments should be developed.

### 5.4. Renewable Energy and Fossil Gas

The updated NDC emphasized Egypt's commitment to move toward more renewable energy as a source of power in many sectors and has taken the first step in investing in large-scale renewable energy projects: "*Installing additional renewable energy (RE) capacities to reach electric power contribution target of 42% by 2035 as per Egypt's Integrated Sustainable Energy Strategy 2035*" [63] (p. 12). However, Egypt is still investing on a larger scale in fossil-fuel-based sources, mainly fossil gas. The government is increasing natural gas consumption in nearly all sectors, for instance by extending the gas pipelines to villages via the Hayah Karima initiative and the rapidly spreading natural gas car stations. While moving toward a lower carbon-intensive fuel is understandable as a bridging strategy, intensive investment in this direction would lock Egypt in a high-carbon pathway with the difficulty of exiting [92].

### 5.5. Strengthening Implementability
#### 5.5.1. Legal Framework

To improve the implementability of the updated NDC, it needs to be enshrined in the comprehensive climate law instead of the fragmentation of laws between sectors and ministries. This law should reflect Egypt's resolution to tackle climate change and mirror the national climate change strategy 2050 issued recently by the Egyptian government [64]. In addition, the next NDC should attach an annex to describe how the participatory approach to designing the NDC is implemented, which Parties took part in defining the main sectors, targets, and measures, and how the final version of the NDC is drafted and agreed on.

#### 5.5.2. Financial Conditionality

The updated NDC did not include unconditional targets. Meeting the defined targets is conditioned on receiving international support. The implementation cost of the updated NDC up to 2030 is estimated at a minimum of USD 246 billion. The updated NDC incorporated a table that distributes this amount across sectors. In this regard, Egypt highly relies on receiving international support according to Article 9 of the Paris Agreement. The NDC stated that Egypt "*is committed to meeting the sustainable development goals under its Vision 2030, and the government has already mobilized significant investments from its local public and private resources*" [63] (p. 30). It is not clear how much or how these resources have been mobilized. Egypt was the first country to issue green bonds as green financing mechanisms. More instruments should be created, such as sustainability-linked bonds, social bonds, dedicated funds for sustainability-focused venture capital, and government grants or procurement programs.

We recommend that A Climate Finance Working Group should be formed to develop a climate finance strategy to align financial flows with the Egyptian climate action and to strengthen the country's Article 6 position, which allows countries to work cooperatively to achieve their emission reduction targets, creating the basis for trading GHG emission reductions in global carbon markets using carbon credit mechanisms [97]. As Egypt was the first country to issue Sovereign Green Bonds in the Middle East and North African region as a green financing mechanism [91], more instruments should follow, such as green loans, sustainability-linked loans, sustainability bonds, and social bonds.

### 5.5.3. Technical Conditionality and Innovation Ecosystem

Another critical condition of implementing the updated NDC is the availability of advanced technology and human resources competencies to tackle climate change in Egypt. The updated NDC affirmed that the Climate Change Central Department (CCCD), the main focal point and coordinator of climate change efforts between all entities, is missing the administrative and analytical competencies required to fulfill its mandates. The updated NDC does not provide an overview of capacity needs or technology needs to implement the mitigation and adaptation actions. This should be clearly defined and submitted as an additional attachment with the successive NDC. Instead of building programs for building capacities inside ministries, the government should adopt an Open Innovation approach by mobilizing all ecosystem actors, particularly specialized research institutes and individual researchers, with the appropriate analytical competencies. There is no better fast-track solution to fill the technological and technical gaps than collaboration with private sectors, big technology companies, specialized research institutes and universities, and international technology transfer agreements.

### 5.5.4. Awareness and Cultural Revolution

The updated NDC is more centralized around measures led by the government on the national or sectoral level. However, nothing is said about changing individual behavior or increasing the awareness of the problem among citizens, consumers, producers, or local municipals. In this line, the government should work and support non-governmental organizations to fill this gap. Changing the culture of consumption and how people deal with natural resources is a role that mass media should play and should be considered in the successive NDC.

## 6. Limitations and Future Research

This study is not without limitations that provide opportunities for further complementary research. First, while our qualitative analysis offers valuable insights into Egypt's climate change efforts, the lack of quantitative methods may limit the depth of our findings. Therefore, future research could complement our qualitative and quantitative analysis to evaluate Egypt's climate change efforts more extensively, for example by using firm-level survey data to investigate how Egyptian firms respond to climate change policies or which factors constitute the most significant barriers to adopting sustainable technologies [98–100].

Second, although our analysis provides a broad view of Egypt's climate change efforts, it primarily focuses on an aggregated national level, lacking deeper insights into specific regions or sectors. Thus, future studies that focus on regional problems and important sectors, such as energy and agriculture, can give more precise and comprehensive policy recommendations, considering specific challenges and opportunities at the regional and sectoral levels [19–21].

Third, the absence of country comparisons in our study may limit our ability to identify best practices and make recommendations based on the relative effectiveness of different policies and measures. Therefore, future research could conduct a comparative analysis with other developing countries to identify best practices and provide recommendations for improvement. This enables us to make more informed recommendations for policy and practice based on evidence from other countries that have faced similar challenges. Moreover, a comparative analysis can help to establish benchmarks and standards for evaluating Egypt's progress in the future and provide insights into how Egypt can learn from the experiences of other developing countries [24,26].

## 7. Conclusions

This paper highlighted the measures Egypt has recently adopted to adapt to climate change impacts in different sectors such as energy, agriculture and irrigation, urban development, and tourism. The efforts undertaken by Egypt demonstrate a proactive approach

towards addressing the challenges posed by climate change and ensuring a sustainable future for the country.

One of the key findings of our research is the significant progress made by Egypt in updating its Nationally Determined Contribution (NDC) in 2022. Compared to its first submission in 2015, the updated version of Egypt's NDC provides a more comprehensive and detailed overview of the country's climate adaptation and mitigation measures. It encompasses a wide range of sectors and highlights quantified targets, concrete actions, and specific steps to be taken.

While acknowledging the commendable progress made, it is important to recognize that certain areas still require attention to further enhance the effectiveness and success of Egypt's climate efforts.

One area that warrants improvement is the mitigation ambition level. Egypt should consider setting more ambitious targets to reduce GHG emissions and actively pursue measures to achieve these targets. By doing so, Egypt can contribute more significantly to global efforts in combating climate change. Transparency is another aspect that needs to be strengthened. It is crucial for Egypt to provide transparent reporting on its progress in implementing the outlined measures and achieving the set targets. This will not only enhance accountability but also facilitate international collaboration and support. Furthermore, Egypt should ensure that its mitigation coverage is more specific and well-defined. This involves clearly outlining the sectors and types of gases that are included or excluded in its mitigation efforts. Such clarity will enable a more focused approach and enable better monitoring and evaluation of the outcomes.

To move towards a cleaner and more sustainable energy future, Egypt should prioritize the installation of renewable energy capacities. By harnessing the potential of renewable sources, such as solar and wind power, Egypt can reduce its dependence on fossil fuels and contribute to a greener and more sustainable energy sector.

In order to enhance implementability, it is essential for Egypt to enshrine its NDC commitments in national climate laws. This will provide a solid legal framework for the implementation of climate actions and ensure continuity and long-term sustainability. Additionally, removing financial and technical conditionalities will help streamline the implementation process and enable smoother progress. Engaging the private sector and civil society in climate change adaptation and mitigation efforts is also crucial. Egypt should actively involve these stakeholders in the planning, implementation, and monitoring of climate initiatives. Their expertise, resources, and innovation can contribute significantly to the success of climate actions. Finally, increasing public awareness and understanding of the negative impacts of climate change and the importance of sustainable practices is vital. Egypt should invest in educational campaigns and outreach programs to educate the general public about climate change and promote behavioral changes that lead to more sustainable practices in daily life.

By undertaking these recommended actions, Egypt can further improve its NDC and make significant strides towards achieving its climate goals. The proactive measures taken by Egypt demonstrate a strong commitment to addressing climate change and ensuring a sustainable and resilient future for the country and its people.

**Author Contributions:** Conceptualization, H.A.; Methodology, H.A. and D.W.; Resources, D.W. and D.M.; Writing—original draft, H.A.; Writing—review & editing, D.W. and D.M. All authors have read and agreed to the published version of the manuscript.

**Funding:** This research received no external funding.

**Informed Consent Statement:** Not applicable.

**Acknowledgments:** The publication of this article was funded by Freie Universität Berlin.

**Conflicts of Interest:** The authors declare no conflict of interest.

## Appendix A

**Table A1.** Detailed NDC analysis following of New Climate Institute' Method [24].

| Element | Intended NDC | Updated NDC | Change |
|---|---|---|---|
| **Scope** | | | |
| **Mitigation** | The NDC gives no absolute emission target level for 2030. The NDC does not present a long-term or net-zero target. | The NDC gives an absolute emission target for 2030. The target covers several sectors. The NDC does not present a long-term or net-zero target. | + |
| *Adaptation* | The NDC presents an adaptation component covering national circumstances, vulnerability, and climate change impact. The NDC contained a section that descriptively outlined several adaptation action packages covering seven sectors. | The NDC presents an adaptation component covering national circumstances and Egypt's vulnerability to climate change impacts. The NDC also reviewed actions taken to implement the first NDC since 2015. The NDC contains a chapter that describes several adaptation measures addressing four sectors, with more details and clearly defined goal-oriented action plans and programs focusing on the main sectors that are more susceptible to climate change in Egypt. In addition, the NDC describes other adaptation measures, such as establishing an early warning system and building resilience for the most vulnerable and marginal regions. | + |
| *Economic diversification* | Not included in the NDC. | The updated NDC provided information on the potential mitigation co-benefits of the adaptation actions, such as the impacts of rainwater harvesting on the use of energy, and the impact of using renewable energy on developing unconventional water sources (e.g., solar desalination). | + |
| *Financing* | Approximately USD 73.04 billion is estimated as a preliminary cost of implementing listed adaptation and mitigation measures. Limited information about financing sources. | NDC indicated that Egypt received USD 10.27 billion from development partners to accelerate the SDG vision. The NDC estimates the total cost of implementing mitigation and adaptation actions at USD 196 billion and USD 50 billion, respectively, and includes a table of cost distribution across sectors. Additionally, the NDC provides more detailed information about financing sources and a separate section about green finance. | + |
| *Technology transfer and capacity building* | The NDC, in one sentence, highlighted the need for the transfer of technology appropriate to the local context and national capacity building. | The NDC emphasizes the importance of technology transfer and capacity building. The document acknowledges that administrative capacities and data management competencies are a challenge, and a supportive legal framework for climate action does not exist. To address these issues, the NDC proposes the use of MRV (monitoring, reporting, and verification) to measure the effectiveness of mitigation and adaptation actions and to scale up measures across Egypt. The NDC also emphasizes the need for climate know-how to be freely available in Egypt to support capacity building, as access to climate-friendly technology is currently limited. | + |

**Table A1.** *Cont.*

| Element | Intended NDC | Updated NDC | Change |
|---|---|---|---|
| **Mitigation ambition** | | | |
| *Country's formulation of the target* | Not included in the NDC. | The NDC does not include a comprehensive target for GHG emissions reduction, but instead provides projections of Business As Usual (BAU) emissions by 2030 for three sectors. The document also quantifies the expected GHG reductions resulting from the implementation of ongoing and planned mitigation projects. | O |
| *Resulting emission level in 2030, excl. LULUCF (land use, land-use change, and forestry)* | Not included in the NDC. | The NDC does not include a comprehensive target for GHG emissions reduction, excl. LULUCF. | O |
| *Conditionality* | Not included in the NDC. | The updated NDC states that a minimum of USD 246 billion in financial resources is required up to 2030, with USD 196 billion needed for mitigation interventions and USD 50 billion for adaptation interventions. The NDC also emphasizes that the implementation of these measures is contingent on receiving adequate international finance through concessional finance and grants. | + |
| *Need for new policies to meet the target* | The NDC included a list of policies and actions to mitigate emissions in the energy and non-energy sectors. | The updated NDC offers a more extensive list of mitigation and adaptation measures compared to the first NDC, providing detailed descriptions of policies implemented since 2015 and future policies. | + |
| *LULUCF and removals* | No quantified emission mitigation targets per sector. | The updated NDC does not provide quantified emission mitigation targets for LULUCF sectors, only details on electricity, oil and gas, and transport. | O |
| *Net-zero target* | Not included in the NDC. | Not included in the NDC. | O |
| *Alignment of NDC with long-term target/ net-zero target* | Not included in the NDC. | Not included in the NDC. | O |
| *Intended use of Article 6* | Not included in the NDC. | The NDC states that Egypt has expressed its interest in participating in emerging international carbon markets that are governed by Article 6 of the Paris Agreement, as a means of voluntary cooperation. | + |
| *Explicit consideration of cost reductions of renewables during the last 5 years* | Not included in the NDC. | No explicit information on cost reductions, but details on the installed amount of renewable energy and increased energy efficiency in the electricity sector. | + |
| *Consideration of ambition of subnational and non-state actors* | Not included in the NDC. | The information provided is limited, with only a few examples given, such as the involvement of private actors in green finance or the "Decent Life" initiative. | + |

**Table A1.** *Cont.*

| Element | Intended NDC | Updated NDC | Change |
|---|---|---|---|
| **Mitigation completeness** | | | |
| *Sector coverage* | **List of mitigation measures:**<br>**1. Energy subsector**<br>- Industry;<br>- Transportation;<br>- Agriculture;<br>- Electricity;<br>- Petroleum.<br>**2. Non-energy sector:**<br>- Agriculture;<br>- Waste;<br>- Industrial processes;<br>- Oil and natural gas. | **Quantified emission reduction targets in three:**<br>- Electricity;<br>- Oil and gas;<br>- Transportation.<br>**List of measures to reduce emissions in the other four sectors:**<br>- Industry;<br>- Building and urban cities;<br>- Tourism;<br>- Waste management. | **(-)** |
| | **Sectors targeted with adaptation measures:**<br>- Water Resources;<br>- Agriculture;<br>- Coastal zones;<br>- Health sector;<br>- Urban development and tourism;<br>- Energy sector. | **Adaptation targeted sectors:**<br>- Water resources and irrigation;<br>- Agriculture;<br>- Coastal zones;<br>Urban development and tourism. | |
| *Gas coverage* | Not included in the NDC. | Three types:<br>- Carbon Dioxide ($CO_2$);<br>- Methane ($CH_4$);<br>- Nitrous Oxide ($N_2O$). | **+** |
| **Mitigation implementation** | | | |
| *Sectoral targets* | The NDC outlines various sector-specific actions to be taken in order to address climate change but provides no quantitative information. | The revised NDC specifies quantified targets for reducing GHG emissions in three sectors (electricity, oil and gas, and transportation). The NDC also includes measures targeting other sectors, although without GHG reduction targets. | **+** |
| *NDC preparation process* | Not included in the NDC. | The NDC states that the planning and implementation of climate measures follows participatory approaches through collaboration among governments, the private sector, civil society, academia, media, the international community, and other concerned stakeholders. | **+** |
| *Clarity of national and sectoral implementation plans* | NDC generally listed different sectoral actions to be taken to mitigate or adapt to climate change. | The updated NDC provides comprehensive information on 35 mitigation and adaptation measures addressing 9 economic sectors.<br>The updated NDC has a clearly defined implementation period and is aligned with different national and sectoral strategies. The Climate Change Central Department (CCCD) is responsible for coordinating climate change efforts between all entities. An MRV has been proposed but not yet fully institutionalized. | **+** |

**Table A1.** *Cont.*

| Element | Intended NDC | Updated NDC | Change |
|---|---|---|---|
| *National climate law(s)* | Not included in the NDC. | Although the NDC enumerates the legal acts that regulate the institutional framework, it lacks clarity regarding whether the NDC or the revised mitigation target is legally binding. As for accountability, the NDC includes quantified targets for three sectors but provides a detailed account of actions aimed at numerous other sectors without specifying quantified targets. | + |
| **Transparency** | | | |
| *Availability of information to estimate resulting emissions level of target* | Not included in the NDC. | "GHG emissions in the base year 2015 were quantified based on data available in Egypt's GHG Inventory submitted to the UNFCCC in 2019 under the First BUR.<br>The modeling of the 2030 projections (BAU and target reductions) was based on an analysis of Egypt's Low Emission Development Strategy (LEDS) utilizing the LEAP software.<br>Explicit calculations, estimations, and projections are reported only for three sectors, and the target for total emissions in 2030 is not reported." | + |
| *Followed the guidance in information to facilitate clarity, transparency, and understanding of NDCs in Decision 4/CMA.1* | Not applicable. | The updated NDC contains a table listing all elements from Annex I to Decision 4/CMA.1, along with information provided for each element. However, certain elements under "scope and coverage," "planning process," and "assumptions and methodological approaches" are stated as being not applicable. | + |
| *Accounting modalities* | Not included in the NDC. | The NDC specifies the specific GWP metrics utilized and mandates that GHG calculations must adhere to the IPCC's guidelines for national GHG inventory. | + |
| **An explanation of why the target is a fair contribution towards the global goal** | Not included. | The NDC provides two justifications for why the revised mitigation target constitutes a fair contribution to the global objective. Firstly, despite Egypt's minimal contribution to global emissions, the NDC highlights the country's susceptibility to the effects of climate change, particularly in relation to the resilience of its coastal and agricultural sectors. Secondly, the NDC notes that the achievement of its present targets is dependent on securing international support for their execution. | + |

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
