# Peer review of "Climate Policy in Developing Countries: Analysis of Climate Mitigation and Adaptation Measures in Egypt"

_sustainability, doi:10.3390/su15119121_

Round 1

Reviewer 1 Report (Previous Reviewer 2)

Dear Author,

You have made a lot of corrections that were advised to you, however, you still need to rethink your introduction and create the part on real literature review.

You are using a lot of technical reports but this is science and you need to use more scientific articles in literature review. It was already recommended to you the following: No theoretical part during the text even compared with other countries. Such article can be useful Kotseva-Tikova, M., & Dvorak, J. (2022). Climate Policy and Plans for Recovery in Bulgaria and Lithuania. Romanian Journal of European Affairs22(2). Art.-5_Maria-Kotseva-Tikova-Jaroslav-Dvorak.pdf (gov.ro) 

Pls, do not ignore it.

Also, you need to think to have normal conclusions.

All the best

Jaroslav

Author Response

Dear Reviewer

Thank you for your valuable feedback on our manuscript. Your input has contributed significantly to improving the quality of our work, and we appreciate your time and effort in reviewing it. 

  • We have restructured our introduction and created a separate theoretical background section. We have included numerous scientific references to enhance our engagement with the extensive scientific discourse on climate change. Additionally, we have included a separate conclusion section.
  • We are now referring to Kotseva-Tikova, M. and Dvorak, J. (2022) to better distinguish the focus of our article.

Sincerely,

Reviewer 2 Report (Previous Reviewer 3)

Dear Authors,

The manuscript has a promising scientific impact on scholars and supports decision-making, as well as the topic in Egypt is very important. 

Although, the paper has been developed. It still needs improvements. Here are my recommendations.

1.       The methodology should be clearly stated in the abstract. The keywords should not be those in the title.

2.       The present type of report is appropriate to be a "Review."

3.       Section 2, the title discusses "Egypt's Economy with Climate Change," it seems rudimental to limit Egypt's economy in the first paragraph (GPD Growth). While the second paragraph, "In 2019, the energy, transport, etc." discusses the emissions. Does the economic aspect within the manuscript's focus? Please demonstrate the rational background behind this attitude. This section's structure should be improved, showing the relation between the introduction and the results.

4.       In the research method, "the intensive desk research investigating published material." A similar study was implemented in Egypt (for instance, and not limited to), and It did not include in the literature review:

 https://doi.org/10.21608/IJISD.2020.73503 It would be better to emphasize the originality of the work. The results should be synthesized and critically analyzed.

 5.       The data seems to be collected and reported rather than critically analyzed and/or interpreted. The coherence of the paper should be improved.

  6.       A conclusion section can be provided that summarizes the findings of the three addressed research questions. 

Author Response

Dear Reviewer

Thank you for your valuable feedback on our manuscript. Your input has greatly contributed to improving the quality of our work, and we appreciate your time and effort in reviewing it. we attach our responses for your kind consideration.

Sincerely,

Reviewer 3 Report (Previous Reviewer 1)

Dear Author,

In the form: Grammar and spelling ( author should revise them carefully).

We recommend revising these points to improve more the manuscript.

1.      Introduction and Theoretical Background : Make the introduction as first section to provide background of this study.

2.      Theoretical Framework (literature) as a second section:  to provide literature review (related cited research papers and studies)

3.      Please, make sure that: Paragraphs are limited to (5 to 6 sentences). Revise all the body of the manuscript.

4.      RESULTS:  (you must add this undersections).

Green Economy and sustainability

Industry 4.0 and Innovation technologies

Climate Adaptation and Mitigation Measures and Achievements (Tourism, Businesses, Mines..)

5.      3.2. Results of NDC Analysis ; these concepts must be analysed;

   Curroptions, worst management, information asymetry,

6.      Conclusion: to summarize the principal results in this paper?Missed

7.      References: Authors should provide for all cited reserch papers the DOI

All references missed the DOI-s.

Revise all the references in this section.

Paper needs more time to be improved and revised,

Good Luck,

Moderate editing of English language is needed.

Author Response

Dear Reviewer,

Thank you for your insightful feedback on our manuscript. Your comments have significantly improved the quality of our work, and we appreciate your efforts in reviewing it. We attach our responses for your kind consideration.

Sincerely,

Round 2

Reviewer 1 Report (Previous Reviewer 2)

Dear Authors,

Thank you for the updated manuscript. You have done a lot of great work.

Some corrections in the list of references still needed. 

You need to correct the citation: Kotseva-Tikova, M., & Dvorak, J. Climate Policy and Plans for Recovery in Bulgaria and Lithuania. Romanian Journal of European Affairs, 2022. 22(2), 79-99

Also, you need to prepare the list of references according to the Sustainability journal requirements. It is must be like this: Snijders, C.; Matzat, U.; Reips, U.D. Big Data: Big gaps of knowledge in the field of internet science. Int. J. Internet Sci. 20127, 1–5.

See also previous example.

All the best

Author Response

Dear Reviewer, 

Thank you for your effort. we corrected the mentioned reference and changed our referencing and citations following the SUSTAINABILITY style. 

Best regards, 

Reviewer 2 Report (Previous Reviewer 3)

The suggestions have been implemented. The article can be accepted.
Thank you 

Author Response

Dear Reviewer, 

Thank you for all comments you provided during the reviewing journey. We are happy that we could provide good responses to your comments. 

Best regards, 

Reviewer 3 Report (Previous Reviewer 1)

Dear authors,

Thank you for your revised draf.

The Reference section : Follow the template SUSTAINABILITY.

Exp: [1]..............................................................

        [2]..............................................................

etc.

All  [1],  [2]....etc. Must be referenced in the body of the manuscript.

Conclusion: to improve more

Good luck.

Moderate revison must be done again.

Author Response

Dear Reviewer, 

Thank you for your comments. We changed all citations and references following the SUSTAINABILITY style and extended our conclusion section. English revision is also conducted by a third party.  

Best regards, 

This manuscript is a resubmission of an earlier submission. The following is a list of the peer review reports and author responses from that submission.

Round 1

Reviewer 1 Report

1. Introduction: The author could improved more the introduction.

2. Egypt: Country Profile  : ( 2.1. Egypt's economy and climate change, 2.2. Egypt's institutional arrangement and policy instruments ) It needs more related studies citations.

3. NDC Assessment Method ;( the following  dimensions: Mitigation - Comprehensiveness - Implementation - Transparency ) shold be more defined and analyzed.

4. Results  : more description is needed ( NDC Implantation Plan -NDC Transparency)

5. Conclusion is missed

6. Limitations is messed.

7. Reference have to be revised and to add recent related studies.

Reviewer 2 Report

Thank you for producing the manuscript which provides data and knowledge on climate policy in Egypt.  There are interesting insights into how the policy is developing in the country. However, despite various strengths, there are some shortcomings:

1. The manuscript is written in a descriptive manner;

2. Unclear the research problem. It seems that if COP27 was not held in Egypt this manuscript never have been prepared. Usually, we solve science problems and not practical ones. 

3. Avoid personal pronouns (we);

4. There is no research aim;

5. What is the research question?

6. No theoretical part during the text even compared with other countries. Such article can be useful Kotseva-Tikova, M., & Dvorak, J. (2022). Climate Policy and Plans for Recovery in Bulgaria and Lithuania. Romanian Journal of European Affairs22(2). Art.-5_Maria-Kotseva-Tikova-Jaroslav-Dvorak.pdf (gov.ro) 

7. The part on methods exists but is still very limited and unclear why these criteria actually the best. Who decide on that?

8. pART 2.2.3 Unclear why these and not other measures were chosen like Green Finance and not Green Economy or Blue Finance as Egypt on the sea short. 

9. What are the conclusions and policy implications?

Thank you

Reviewer 3 Report

Dear Authors,  

 The manuscript provides a qualitative approach to examine the institutional and legal frameworks in terms of tackling climate change from a holistic perspective. Although the paper discusses a critical topic supporting Egypt's decision-maker and scientific community, it requires improvements before acceptance.

1.       The methodology is not clear, and it should be clearly indicated. What are and why are the considered factors investigated? It seems they were selected based on availability. For example:

-          Section 2.2.3. indicated, "The following will briefly overview the most significant measures and achievements in energy, agriculture and livelihood,  urban development, and green finance. "What is the rational background for selecting these items?

-          Please provide a summary of the obtained methodology, Reference Villafranca Casas et al., 2021, and in line with the appendix. Considering that the study focuses on developing countries. Thus, what is the critical analysis of the methodology of Villafranca Casas et al. as a methodology in a developed country?

-          Please define what the signs mean in the third column.
Please define the abbreviations once they appear in the manuscript, such as Land Use, Land-Use Change, and Forestry (LULUCF); also, does it fit the Egyptian context?

-          Adding a visual summary of the methodology might be beneficial. 

2. Page 3, Section 2: "On the other hand, Egypt has a strong demographic pyramid with a base of youth; approximately 60% of the population is under the age of 30, and 40% is between the ages of 10 and 29 (UNFPA, 2016). This helped the country to show resilience against the COVID-19 pandemic; however, climate and innovation policies are needed to unleash these youth's innovation and entrepreneurial competencies and mobilize them to tackle climate change impacts." 

- What is the relevance of the demographic characteristics and Covid-19 pandemic to the study context? Then you mentioned: "climate and innovation policies are needed to unleash these youth's innovation and entrepreneurial competencies and mobilize them to tackle climate change impacts." This is a bit surprising to provide insights and analysis in the background section without grounding it. 

- Please elaborate on what "Ayesh in Arabic, which means living," conveys to the study context. 

- Page 3. "In this context, climate change impacts cannot be addressed separately from discussing Egyptian people's social and economic political rights." What is the correlation of socioeconomic rights to climate change strategies? The study reviews the policies. So please, if it focuses on this, please review the relevant policies in line with climate change. For instance, what about human rights strategy with climate change? The entire section looks surprising.

3. Page 6, "The Upper Egypt Local Development Program in southern Egypt contains policies to boost municipal solid waste management, conduct climate risk assessments, and develop local climate action plans to address risks and mitigate people's exposure to them and the local economy (World Bank, 2023f)."

Please clarify in which part of the reference (World Bank, 2023f) this statement was indicated. 

It seems that. The exact sentence was extracted as it is from another reference
https://www.worldbank.org/en/news/opinion/2022/04/19/-egypt-acting-against-climate-change-for-a-healthier-more-prosperous-future 

4.       Page 1: "Egypt's recent actions." In terms of what? Sectorial level, policies, etc. 

5. The manuscript in its current form is suitable as a review paper, not a research article.

6. Page 2, Section 2, has no significant contribution. It could be shortened. However, you mentioned on page two, second paragraph, "Section two presents a country profile demonstrating why Egypt is too susceptible to climate change." It does not make sense. Also, Figure one is not relevant to the study content.

7. Please define who are the "international actors" mentioned on pages 2 and 9. To what extent can they affect the local context?

8.       Page 3, section 2.1: "Emissions have trended upward between 1990 and 2019, both in absolute terms (163% increase) and per capita terms (47% increase) (WRI, 2023)." Does The numbers reference is for this emission globally? Or local context.  

9. Please show the juxtaposition of this study with similar previous studies discussing/evaluating the Egyptian climate actions in line with the considered factors in this report. For example but not limited to: 

https://doi.org/10.3390/w13121715 

https://doi.org/10.3390/land11122237 

https://doi.org/10.3390/w13040412 

The results should be synthesized and critically analyzed. Please work on emphasizing the originality of this work.